# Hydrogen Inhalation Reduces Lung Inflammation and Blood Pressure in the Experimental Model of Pulmonary Hypertension in Rats

**DOI:** 10.3390/biomedicines11123141

**Published:** 2023-11-25

**Authors:** Tatyana Kuropatkina, Dmitrii Atiakshin, Fedor Sychev, Marina Artemieva, Tatyana Samoilenko, Olga Gerasimova, Viktoriya Shishkina, Khaydar Gufranov, Natalia Medvedeva, Tyler W. LeBaron, Oleg Medvedev

**Affiliations:** 1Department of Pharmacology, Faculty of Medicine, Lomonosov Moscow State University, Lomonosovsky Prospect 27-1, 119991 Moscow, Russia; 0sylphide0@gmail.com (T.K.); artemievamm@my.msu.ru (M.A.); haidar@formed.ru (K.G.); 2Research Institute of Experimental Biology and Medicine, N.N. Burdenko Voronezh State Medical University, Moskovsky Prispect, 185, 394066 Voronezh, Russia; atyakshin-da@rudn.ru (D.A.); antailkka@mail.ru (T.S.); stavro7@mail.ru (O.G.);; 3Research and Educational Center for Immunophenotyping, Digital Spatial Profiling and Ultrastructural Analysis Innovative Technologies, People’s Frendship University of Russia, Miklukho-Maklaya St. 6, 117198 Moscow, Russia; 4Faculty of Biology, Lomonosov Moscow State University, Leninskie Gory 1-12, 119234 Moscow, Russia; sychev.sychevfedor2018@gmail.com (F.S.); namedved@gmail.com (N.M.); 5Department of Kinesiology and Outdoor Recreation, Southern Utah University, Cedar City, UT 84720, USA; tylerlebaron@suu.edu; 6Molecular Hydrogen Institute, Cedar City, UT 84720, USA; 7Laboratory of Experimental Pharmacology, National Medical Research Center of Cardiology Named after Accademician Chazov E.I., Akademika Chazova St. 15a, 121552 Moscow, Russia

**Keywords:** molecular hydrogen, selective antioxidant, hydrogen inhalations, ROS, pulmonary hypertension, inflammation, mast cells, tryptase, TGF-β

## Abstract

Hydrogen has been shown to exhibit selective antioxidant properties against hydroxyl radicals, and exerts antioxidant and anti-inflammatory effects. The monocrotaline-induced model of pulmonary hypertension is suitable for studying substances with antioxidant activity because oxidative stress is induced by monocrotaline. On day 1, male Wistar rats were subcutaneously injected with a water–alcohol solution of monocrotaline or a control with an only water–alcohol solution. One group of monocrotaline-injected animals was placed in a plastic box that was constantly ventilated with atmospheric air containing 4% of molecular hydrogen, and the two groups of rats, injected with monocrotaline or vehicle, were placed in boxes ventilated with atmospheric air. After 21 days, hemodynamic parameters were measured under urethane narcosis. The results showed that, although hydrogen inhalation had no effect on the main markers of pulmonary hypertension induced by monocrotaline injection, there was a reduction in systemic blood pressure due to its systolic component, and a decrease in TGF-β expression, as well as a reduction in tryptase-containing mast cells.

## 1. Introduction

Dihydrogen (H_2_) is a colorless, non-toxic, tasteless, and odorless gas. Inhalation of molecular H_2_ has been shown to reduce perfusion deficits in cerebral ischemia due to its ability to neutralize highly reactive hydroxyl radicals and the reactive form of nitric oxide, peroxynitrite (ONOO^−^), which has a marked cytotoxic effect [1,2,3].

Molecular H_2_ is considered to be one of the most selective antioxidants, as demonstrated by Ohta et al. [3,4]. They demonstrated that hydrogen, while significantly reducing the levels of ^•^OH in cells, does quench important cellular levels of various signal oxidants including O_2_^−^, H_2_O_2_, and NO^•^. This is because the high oxidative activity of ^•^OH allows this radical to react with the hydrogen molecule, whereas the much lower oxidative activity of O_2_^−^, H_2_O_2_, and NO^•^ is insufficient to react with H_2_ [2,3,4]. As a result, hydrogen can reduce oxidative stress and correct the redox status of cells without altering the physiological processes occurring in cells [5]. This medical gas is also a substance with no dose limitations and no known toxic effects [4,6,7].

In some reports, the properties of hydrogen are not limited to the ability to scavenge the hydroxyl radical, but also to inhibit pro-inflammatory and inflammatory cytokines, such as interleukins 1 and 6, tumor necrosis factor (TNF-α), etc. Thus, H_2_ can also reduce the expression of pro-apoptotic factors including B-cell lymphoma-2 (BCL-2)-associated X-protein (BAX), caspases 3, 8, and 12, and activate anti-apoptotic factors such as the anti-apoptotic proteins Bcl 2 and Bcl XL [1,8,9]. Accordingly, H_2_ can be considered as a molecule that regulates intracellular signaling pathways that coordinate the biochemical processes of cells in the body [10].

Several studies confirm that oxidative stress plays a key role in the pathogenesis of cardiovascular diseases [11], including pulmonary hypertension (PH), which is characterized by endothelial dysfunction, increased right ventricular (RV) pressure, and hypertrophy [12]. Increased reactive oxygen species (ROS), altered lung redox state, and elevated RV pressure have been demonstrated in several models of PH, including monocrotaline-induced PH [13]. The generation of ROS in these models occurs mainly due to the activity of NADPH oxidase, xanthine oxidase, and endothelial NO synthase. As the disease progresses, circulating monocytes and monocytic precursor cells from the bone marrow migrate and accumulate in the pulmonary vasculature. Once established, these inflammatory cells generate ROS and secrete mitogenic and fibrogenic cytokines, which induce cell proliferation and fibrosis in the vessel wall, leading to progressive vascular remodeling [13].

Monocrotaline-induced PH (MCT-PH) is the most commonly used experimental model. Endothelial cells treated with MCT show pronounced disturbances in intracellular membrane processes affecting several cell membrane proteins [14]. Cell damage by MCT pyrrole results in the increased synthesis of pro-inflammatory cytokines such as IL-1, IL-6, and TNF-α [15,16,17]. It also increases the secretion of a major fibrogenic factor, connective tissue growth factor (CTGF) [18], which contributes to scarring in the lungs, and the hypertrophy of the pulmonary artery and right ventricular heart tissue.

In MCT PH, the hypertrophy of the medial elastic and muscular pulmonary arteries is not accompanied by the proliferation and muscularization of the small interstitial arteries [13]. The MCT-induced PH model provides repeatable, reproducible results and implies a number of changes that could theoretically be influenced by molecular hydrogen. The aim of this work was to investigate the effect of breathing 4% hydrogen in ambient air on the symptoms of a monocrotaline-induced model of pulmonary hypertension in male Wistar rats. The blood pressure, histological changes, and the degree of inflammation in lung tissue were evaluated.

## 2. Materials and Methods

Experiments were performed on 2-month-old male Wistar rats (180–220 g body weight). Manipulations with the animals were performed according to the principles of Council Directive 86/609/EEC. The protocol of the experiments was approved by the Ethics Committee of Faculty of Biology, MSU (Approval 113-G, 19.06, 2020). The animals were obtained from the vivarium of the Research Institute of General Pathology and Pathophysiology (Moscow, Russia). The rats were kept under 12 h daylight conditions with free access to water and food, and humidity and temperature were controlled. The adaptation period after transport was at least 7 days. The rats were then weighed, and their systolic blood pressure was measured twice using the tail-cuff plethysmography technique.

On day 1, two groups of animals received a single subcutaneous injection of monocrotaline (MCT) (60 mg/kg in 60% ethyl alcohol) (Sigma Aldrich, Darmstadt, Germany). The control group received only a solvent for MCT (60% ethyl alcohol) subcutaneously, i.e., it was the control for the MCT effect.

Animals receiving the MCT injection were divided into two groups: those inhaling room air (MCT control) and those inhaling a mixture of room air and 4% hydrogen (MCT-H_2_). The inhalations were continuous except for the period of cage cleaning (1 h/3 days). Animals inhaled either air or air containing 4% H_2_ for 21 days. There were four rats in each cage, and two cages in each group (control group, MCT-group, and MCT + H_2_ group).

### 2.1. Experimental Setup

Rats were placed in 130-liter plastic containers (SAMLA 203.764.41, Inter IKEA Systems). Three containers were used. Two T2 cages for laboratory animals were placed inside each container. The rats were housed in groups of 4 in each individual cage. So, there were two cages (8 rats) in each group: control, MCT-group, and MCT + H2 group. To ventilate the containers, a Linear Air Pump (SPP-15GAS, Hiblow XP 40, Techno Takatsuki, Japan) was used. Molecular hydrogen was produced by the generator (Pioneer, OOO, Vodorodpomogaet, Russia). Figure 1 shows a schematic of the experimental setup.

The actual hydrogen and air flow rates were controlled by the rotameter valves (LZB-3, LZM-4T, China) and kept at rates of 0.15 L/min and 4 L/min, respectively. It created the 3.5–4.0% of the hydrogen in the air that was used for cage ventilation.

### 2.2. Measurement of Hemodynamic Parameters In Vivo

Animals were anaesthetized intraperitoneally with urethane (aqueous solution, 1.2 g/kg, 0.6 g/mL). Mean blood pressure (MBP) and right ventricle systolic pressure (RVSP) were evaluated directly using the Statham blood pressure transducer (Statham Instrument Inc., Los Angeles, CA, USA), an operational amplifier, and the L-Card E14–140 multichannel analog-to-digital converter (L-Card, Moscow, Russia). For that purpose, the PE10 catheter was placed into the femoral artery, and the PE 50 catheter (Medsil, Moscow, Russia) was inserted into the RV through the right jugular vein. The RVSP rate was used to evaluate the severity of pulmonary hypertension.

### 2.3. Morphometric Measurement

Euthanasia was performed in urethane-narcotized rats via decapitation. After euthanasia, the heart was taken out and washed in a saline solution, the atrium was cut out, and the left ventricle was separated from the right ventricle and the interventricular septum. The assessment of the RV hypertrophy degree was estimated relative to the sum of the left ventricular (LV), and interventricular septum (S) mass (RV/(LV + S)) is reported in conventional units. The lung mass was also measured.

### 2.4. Morphological Lung Tissue Investigation

Lung samples were fixed in 10% neutral formalin and subjected to standard sample preparation procedures. The resulting 5 µm thick cross-sections were stained with hematoxylin and eosin, and May–Grunwald and Giemsa solutions were used according to Mallory’s method to identify fibrotic changes in the lung parenchyma. Mast cell tryptase expression was assessed with immunohistochemistry using a mouse monoclonal antibody to mast cell tryptase (#ab2378, dilution 1:2000, Abcam, UK) and lung fibrosis was assessed using a rabbit monoclonal antibody to TGF-beta1 [EPR21143] (#ab215715, dilution 1:500, Abcam, UK).

All histological sections were analyzed blindly using a Zeizz Imager.A2 microscope, Zen 3.0 Light Microscopy Software Package. The microplates were analyzed in 30 fields of view at a magnification of ×200.

Planimetric analysis to determine the area of the extracellular matrix of lung connective tissue was determined using open-source software for digital pathology image analysis QuPath [19] with further calculations of the relative content.

### 2.5. Statistical Analysis

Data collected during the experiment were plotted and presented as mean ±standard deviation (M ± SD). Statistical analysis of the data was performed using Statistica 12.0 (Statistica Inc., Palo-Alto, CA, USA) and GraphPad Prism 8.0 software. The normality of distribution was tested using the Shapiro–Wilk test. One-way ANOVA was used to determine whether there was a statistically significant difference in one factor between the means of three or more independent groups. Two-way ANOVA was used to determine the simultaneous effect of group and exposure duration, and to assess the interaction between these factors. The Kruskal–Wallis test was used to compare ranks. Statistical outliers were excluded using the ROUT criterion with Q not >1%. Qualitative data were described as absolute frequencies (*n*) and relative frequencies (%). Data were considered statistically significant if the confidence level did not exceed *p* < 0.05.

## 3. Results

### 3.1. Effects on Blood Pressure

MCT induced significant elevations in RSVP and relative RV mass with no significant changes in heart rate. The inhalation of 4% H_2_ in the air did not show any changes in the main symptoms of MCT PH. Specifically, as illustrated in Figure 2A, there was no statistically significant difference (*p* > 0.05) between the groups. The value of RVSP in the MCT control group was 57 ± 9 mm Hg and in the MCT-H_2_ group it was 58 ± 9 mm Hg, which were both significantly higher compared to the control animals (*p* < 0.05), which averaged 37 ± 5 mm Hg (Figure 2A). There were also no differences in HR between groups (Figure 2B).

MCT injection significantly increased the relative RV mass (*p* < 0.05) compared to control groups, which confirms the development of PH in all groups (Table 1). However, 4% H_2_ inhalation had no effect on the relative RV mass compared to the MCT-only group (*p* > 0.05) (Table 1).

The mean arterial pressure was significantly lower (82 ± 7 mm Hg) in the group breathing atmospheric air containing 4% H_2_ compared to either the MCT-Control (95 ± 7 mm Hg) and control (93 ± 9 mm Hg) groups, *p* < 0.05 (Figure 3A). The observed difference was shown to be due to a decrease in its systolic component: 120 ± 15, 121 ± 9, and 107 ± 8 mm Hg for the control, MCT-Control, and MCT-H_2_ groups, respectively (Figure 3B). Moreover, there were no differences (*p* > 0.05) in diastolic blood pressure among the three groups (Figure 3B).

### 3.2. Morphological Analysis of Lung Tissue

Morphological analysis of the left lung tissue sections stained with hematoxylin and eosin in the MCT-Control groups revealed pathological changes inherent to pulmonary hypertension (Figure 4). In all animals examined within the MCT-Control group, there was evidence of arterial medial hypertrophy. The presence of thrombus, which is associated with the development of pulmonary fibrosis, as well as arteriolar luminal narrowing, was observed. In addition, signs of chronic inflammation and perivascular and peribronchial lung tissue were diffusely infiltrated with immunocompetent cells.

Alveolar macrophages accumulated in the lumen of many alveoli (Figure 4C). The perivascular infiltration of lymphocytes forming lymph nodes was seen (Figure 4D). The walls of the respiratory acinus and some airways developed edema (Figure 4D). Fibroblast proliferation and plasma cell migration occurred in many limited areas of the lung stroma (Figure 4E). Changes in the wall structure of many arterioles were accompanied by significant luminal narrowing and muscular sheath hypertrophy (Figure 4F). Giemsa staining allowed for the quantitative analysis of the metachromatic population of mast cells (MCs). The highest MC activity with signs of degranulation was observed in areas of developing fibrosis and around stenosed vessels (Figure 5). The accumulation of MCs occurred both in the adventitia of large vessels and airways and in the airways (Figure 4G,H), which was rarely observed in control animals. At the same time, mast cells were often in close association with other immunocompetent cells, in particular neutrophils and eosinophils (Figure 4G).

Staining using the Picro Mallory method revealed vascular changes proceeding against the background of MCT action: in large branches of the pulmonary artery, there was the remodeling of fibrous connective tissue components in the adventitial sheath with the formation of fibrosis signs (Figure 5). Fibrotic changes were seen in the lung parenchyma with irregular density, resulting in the growth of collagen fiber bundles, reduction in the lumen of respiratory structures, and thickening of the vascular channel wall (Figure 5F). In some loci of the airways, there was a significant increase in connective tissue and collagen fiber bundles (Figure 5G,H).

The integral assessment of connective tissue content in the lungs on scanned micro-documents showed an increase in both absolute and relative values of this index (Table 2). The expression of tryptase by MCs was significantly higher compared to the control group with signs of varying degrees of degranulation (Table 3).

TGF-β secretion by alveolar macrophages and type II alveolar cells was high compared to the control and MCT-H_2_ groups (Figure 6). With a general increase in the number of TGF-β+ cells, their uneven distribution in the respiratory part of the lung was observed with the formation of a preferential accumulation in certain loci of the acinus (Figure 5J).

The morphological analysis of samples from the MCT-H_2_ group showed that the structures of the acinus acquired a classical shape; the alveoli had a cellular appearance and slightly thickened walls with full blood capillaries in the interstitium (Figure 4I). Signs of interstitial edema decreased, but pneumocytes with signs of hypertrophy persisted (Figure 4I). The MC response showed both a quantitative and functional decrease compared to the MCT control group. Analysis of tryptase expression by MCs showed a tendency to decrease with the prevalence of functional forms without signs of degranulation (compared to the MCT control group) (Table 2, Figure 4G–L). Analysis of micro-preparations stained using the Picro Mallory method did not reveal any significant degree of fibrosis (Figure 5K–M, Table 3). However, increased connective tissue content persisted in some alveoli (Figure 5L,M). The phenomena of the adventitial wall remodeling of large branches of the pulmonary artery with the formation of fibrous nodules were noted. TGF-β secretion was significantly reduced in the MCT-H_2_ group compared to the MCT control group and was demonstrated by single cells in alveolar structures (Figure 5N and Figure 6).

Histochemical analysis showed that the number of MCs (including those containing tryptases) in the MCT-H_2_ group was on average 30% lower than in the MCT-Control group (Table 2), *p* < 0.05.

## 4. Discussion

Considering that molecular hydrogen has antioxidant and anti-inflammatory properties and may have therapeutic effects on the pathogenesis of various diseases [1,2,3,4,5,6], the present study investigated the effects of inhalation of 4% H_2_ mixed with atmospheric air on the symptoms of MCT-PH development, morphology, and inflammatory markers in the lungs of experimental rats. Despite previous studies showing the successful application of hydrogen in the development of PH [20,21,22], our study did not show similar effects of 4% H_2_ in air breathing on the main symptoms, RVSP, and RV hypertrophy development. This may be due to the fact that, in the previously published articles, the authors used hydrogen-rich water or intraperitoneal injections of H_2_-rich saline, whereas in our study we used continuous inhalation of the 4% H_2_. It is known that the kinetics of H_2_ are different for different routes of administration [23,24]. Moreover, it has also been reported that, whereas either drinking hydrogen water or intermittent inhalation of 2% H_2_ gas had favorable effects in a rat model of Parkinson’s disease, continuous exposure to 2% H_2_ gas had no significant favorable effects [25]. Additionally, it is possible that a higher concentration of hydrogen gas would have elicited a more noticeable effect. Indeed, some studies use up to 66.67% inhalation of hydrogen gas such as with COVID-19 patients [26]. However, we chose 4% H_2_ because clinical research has shown therapeutic effects using 2–4% H_2_ [27] and H_2_ gas is not flammable blow a 4.6% concentration.

However, despite the lack of influence on the main parameters of PH development, we found a protective effect on the morphological changes in the lungs caused by the development of MCT PH and the level of inflammatory process in them. Perhaps the discrepancy between the attenuation of pathological morphological changes and the null effects on RVSP may be explained by the small sample size, relatively short duration of the study, and other unknown mechanisms that contribute to RVSP. However, in our data, it is clear that H_2_ exerted a protective effect on the lungs. Indeed, the histological data indicate that the use of continuous inhalation of 4% molecular H_2_ reduced the degree and frequency of lung tissue fibrosis.

This effect may be explained in part by a marked decrease in the production of transforming growth factor-β (TGF-β), which is consistent with previous studies by other authors. TGF-β is a multifunctional mediator that regulates proliferation, differentiation, apoptosis, adhesion, and migration in various cells such as macrophages, activated T and B cells, immature hematopoietic cells, neutrophils, and dendritic cells. TGF-β1 is expressed in endothelial, hematopoietic, and connective tissue cells. It is now generally accepted that TGF-β/Smad signaling is an important pathway for fibrogenesis such as renal fibrosis, hepatic fibrosis, and pulmonary fibrosis. TGF-β signaling is increased in macrophages, airway epithelium, smooth muscle cells, and fibroblasts in various lung diseases. TGF-β stimulates myofibroblast differentiation as an aberrant response to injury, which exacerbates progressive pulmonary fibrosis [28].

Tao et al. showed that hydrogen can inhibit the production of pulmonary TGF-β, which is involved in the reprogramming of gene expression during epithelial-mesenchymal transition (EMT) and provokes chronic inflammation, excessive extracellular matrix (ECM) deposition, and the scarring of lung tissue [29]. In addition, the activation of inflammation and an increase in the number of MCs in lung tissue, which also contribute to lung tissue remodeling, were observed against the background of monocrotaline action. During degranulation, MCs release a wide range of mediators that can be classified into three groups, including preformed mediators (e.g., histamine, tryptase, and chymase), de novo mediators including prostaglandin (PG) D2, leukotriene (LT) B4, and LTD4, and a long list of cytokines and growth factors such as tumor necrosis factor (TNF)-α, TGF-β, vascular endothelial growth factor (VEGF), granulocyte-macrophage colony-stimulating factor (GM-CSF), IL-10, IL-8, IL-5, IL-3, and IL-1 [5,6]. Future research should measure these other mediates in order to elucidate the molecular mechanism of molecular hydrogen. In fibrotic lung disease, the interaction between MCs and fibroblasts contributes to a profibrotic environment in which fibroblasts support MC survival and proliferation by producing stem cell factor (SCF). In turn, the MC-derived chymase activates latent TGF-β1, which mediates fibroblast differentiation into myofibroblasts [30].

Through secretome components, MCs are intimately involved in the genesis of adaptive and pathological conditions and represent not only an informative marker of disease progression but also a promising therapeutic target. Specific MC proteases, such as tryptases, are of great importance [31].

The number of tryptase-secreting MCs is known to increase and correlate with the severity of pulmonary hypertension and pulmonary vascular remodeling [32]. The inhalation of 4% hydrogen resulted in a decrease in the number of MCs, including those containing tryptase, confirming its anti-inflammatory properties. In addition, monocrotaline has been shown to activate inflammation and increase the number of MCs in lung tissue, which also contributes to lung tissue remodeling. It is evident that the antifibrotic effects of molecular hydrogen could be largely based on the reduction in both the number and secretory activities of TCs, since they play an active role in collagen fibrillogenesis [33]. The number of tryptase-secreting MCs is known to increase and correlate with the severity of pulmonary hypertension and pulmonary vascular remodeling [32]. The inhalation of 4% hydrogen resulted in a decrease in the number of MCs, including those containing tryptase, confirming its anti-inflammatory properties.

Although we did not observe an effect of H_2_ on symptoms of MCT PH development such as increased RVSP and RV hypertrophy, the effect of H_2_ exposure on the magnitude of systemic blood pressure was demonstrated. Our study reported a significant reduction in mean and systolic blood pressure in the MCT-H_2_ group compared with hypertensive and normotensive controls. A randomized, placebo-controlled study by Liu et al. showed similar results. The authors included 60 patients with diagnosed arterial hypertension, aged 50–70 years, who were exposed to inhalation of room air or its mixture with hydrogen (0.2–0.4%) for 4 h a day for 2 weeks [34]. It was found that systolic blood pressure decreased significantly after hydrogen inhalation compared to baseline, which was not observed in the placebo group. The authors suggest that the observed effect is due to the fact that the use of hydrogen contributed to a significant reduction in angiotensin II, cortisol, aldosterone, and their plasma renin ratio compared to baseline levels [34].

A similar result was shown in the work of Nakayama et al., where using a hemodialysis solution with dissolved H_2_ was able to improve post-dialysis MBP by reducing the systolic component, but the authors did not provide mechanisms for achieving this effect [35].

Our results indicate that 4% H_2_ in the breathing air causes a decrease in mean BP due to its systolic component. This may be due to the effects of hydrogen on the regulatory effects of the autonomic nervous system by decreasing sympathetic nerve activation [36], promoting changes in the elasticity of large arterial vessels by enhancing nitric oxide production, [37] as well as to its antioxidant, anti-inflammatory, and antifibrotic effects [28,34].

Although the favorable effects observed in our study are provocative, additional research needs to be conducted before making stronger conclusions. The study was limited by a small sample size (*n* = 5) of rats, all of which were male. There may also be some sex differences as suggested by other research on H_2_ [38] that should be investigated. Moreover, as mentioned, different routes of administration as well as different concentrations, frequencies, and durations of treatment need to be explored in order to determine the optimal method of administration. Furthermore, more mechanistic investigation is needed in order to determine the mode of action and primary target(s) of molecular hydrogen in this disease. It would also be useful to monitor the levels of TGF-β1 positive cells in other tissues in order to elucidate its pleiotropic effects.

Nevertheless, these results in combination with previously published results indicate that molecular hydrogen may be a useful therapeutic agent to combat pulmonary hypertension. Accordingly, additional mechanistic and clinical investigation is warranted.

## Figures and Tables

**Figure 1 biomedicines-11-03141-f001:**
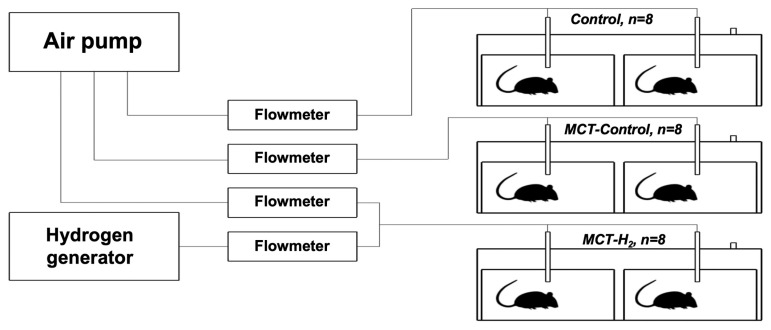
Schematic representation of the experimental setup.

**Figure 2 biomedicines-11-03141-f002:**
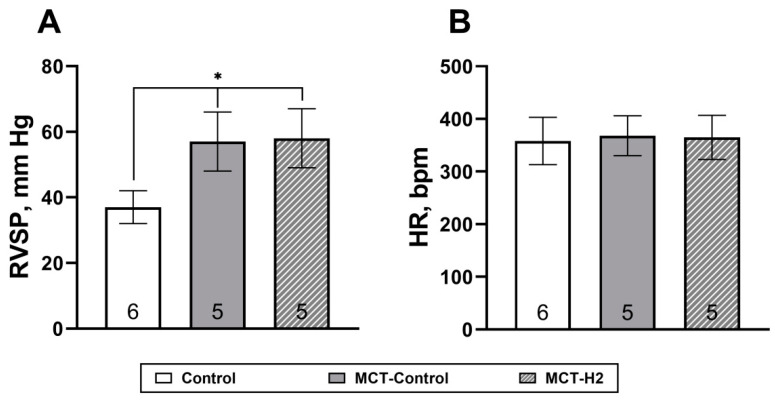
(**A**) Right ventricular systolic pressure (RVSP). (**B**) Heart rate (HR) on day 21 of the experiment. * Control vs. MCT-Control, MCT-H_2_, *p* < 0.05, one-way NOVA.

**Figure 3 biomedicines-11-03141-f003:**
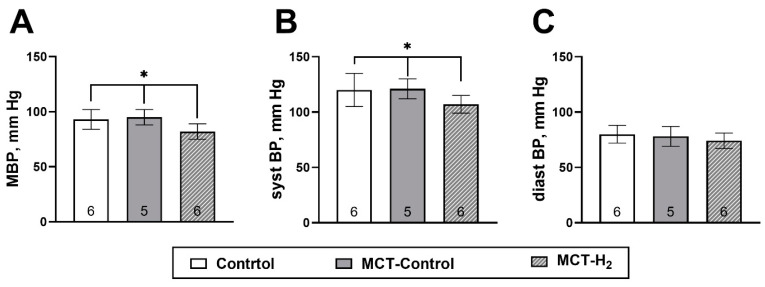
Mean arterial blood pressure (**A**), systolic (**B**), and diastolic (**C**) blood pressure on day 21 of the experiment * MCT-H_2_ vs. control, MCT-Control, *p* < 0.05, one-way ANOVA.

**Figure 4 biomedicines-11-03141-f004:**
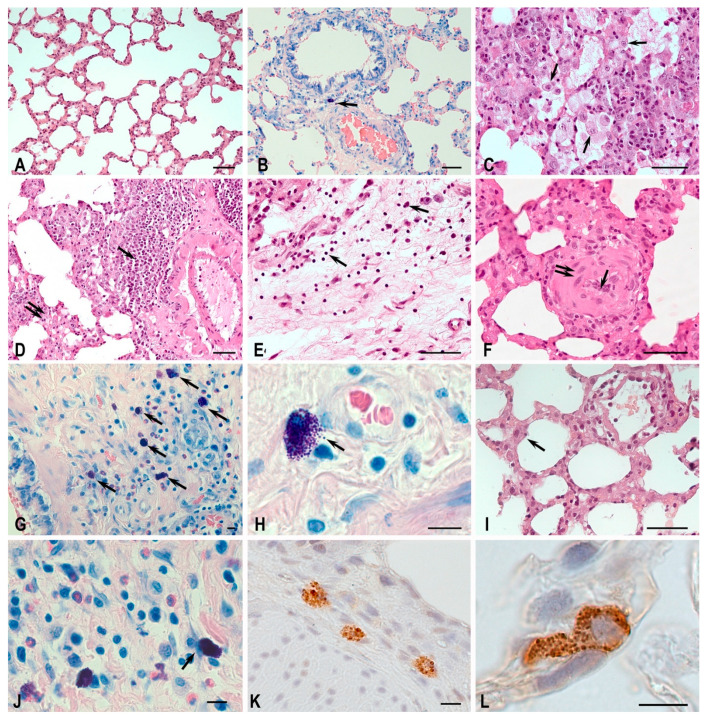
Histological characteristics of lung structures and mast cells of Wistar rats. Methods: hematoxylin and eosin staining (**A**,**C**–**F**,**I**), May–Grunwald and Giemsa solution staining (**B**,**G**,**H**,**J**), immunohistochemical detection tryptase of MCs (**K**,**L**). (**A**,**B**) Control group. (**A**) In the respiratory part of the lung, elongated alveolar passages pass into thin-walled alveoli; single macrophages are found in interalveolar septa. (**B**) Lung MCs were characterized by sparse representation, localized perivascularly and in the bronchial wall, mostly without, or with a weak degree of degranulation (arrow). (**C**–**H**) MCT-Control group. (**C**) Accumulations of alveolar macrophages in the lumen of alveoli (arrow). (**D**) Perivascular infiltration by lymphoid tissue with nodule formation (arrow) against the background of pronounced interstitial edema (double arrow). (**E**) Remodeling of connective-tissue stroma with fibroblast proliferation foci (spindle-shaped cells), lymphocytic-plasmocytic representation is observed (arrow). (**F**) Structural and functional changes in arteriolar wall and intima obliteration (arrow) with media hypertrophy (double arrow) are observed. (**G**) MC accumulation (arrow) in the bronchial wall in association with neutrophilic-eosinophilic granulocytes. (**H**) Active degranulation of MCs (arrow). (**I**–**L**) MCT-H_2_ group. (**I**) Less pronounced edema of interstitial septa, and pneumocytes with signs of hypertrophy (arrow). (**J**) MCs with weak degranulation, and colocalization with a fibrocyte (arrow). (**K**,**L**) Tryptase-positive MCs in lung tissue. Scale bar: (**A**–**F**,**I**) 50 µm, the rest 10 µm.

**Figure 5 biomedicines-11-03141-f005:**
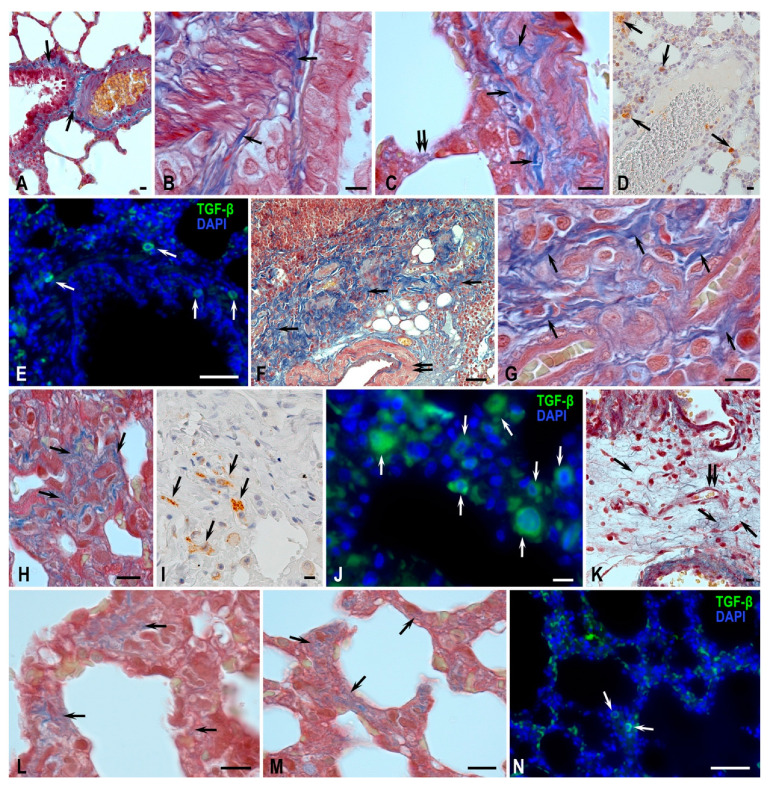
Morphological analysis of the extracellular matrix remodeling process in the lungs of Wistar rats. Methods: Picro Mallory method of staining (**A**–**C**,**F**–**H**,**K**–**M**) and immunohistochemical detection of TGF-β (**D**,**E**,**I**,**J**,**N**). (**A**–**E**) Control group. (**A**–**C**) Thick bundles of collagen fibers in the stroma of bronchi and pulmonary vessels (indicated by arrow); only thin fibers are detected in the respiratory compartment (double arrow). (**D**,**E**) Representation of perivascularly located TGF-β-positive cells (arrow). Low content of TGF-β-positive cells localized predominantly in the stroma of a large bronchus (arrow). (**F**–**J**) MCT—control group. (**F**) Marked fibrotic changes in the form of overgrowth of densely packed connective tissue fibers in the pulmonary parenchyma (arrow), decreased airiness of the pulmonary tissue, and thickening of the vascular wall (double arrow). (**G**,**H**) Significant increase in connective tissue and collagen fiber bundles in some loci of the respiratory lung (arrow). (**I**) Scattered groupings of cells secreting TGF-β (arrow) surrounded by connective tissue fibers and thin-walled vessels. (**J**) Aggregation of large numbers of TGF-β-positive cells in some loci of respiratory lung acinuses (arrow). (**K**–**N**) MCT—H_2_ group. (**K**) Pattern of granulation tissue as a variant of resolution of the process of sclerosing of the intervascular stromal space, thin loose connective tissue fibers (arrow) with elements of neovasculogenesis (double arrow). (**L**,**M**) Low content of fibrous component of extracellular matrix in the structures of respiratory acinus (arrow). (**N**) Low content of TGF-β-positive lung acinus cells (arrow). Scale bar: (**E**,**F**,**N**) 50 μm, others 10 μm.

**Figure 6 biomedicines-11-03141-f006:**
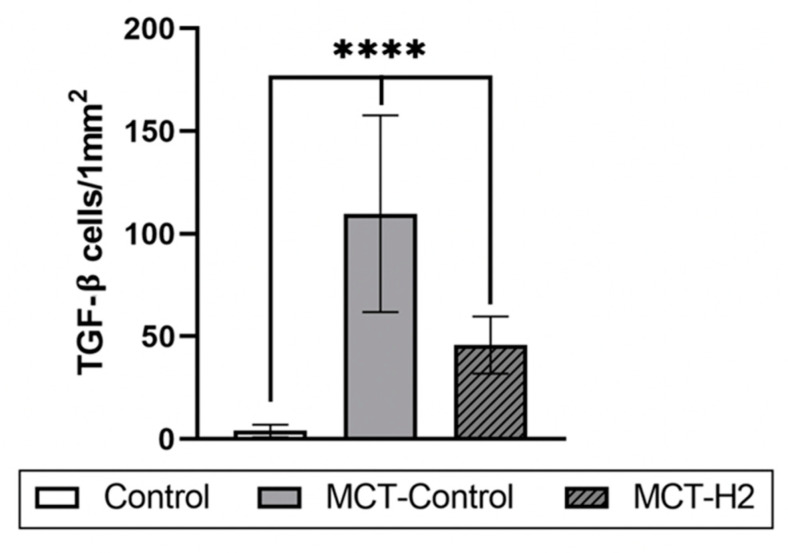
The mean number of TGF-β-positive cells in the lung tissue on day 21 of the experiment. **** Control vs. MCT-Control, MCT-H2, MCT-H2 vs. MCT-Control, *p* < 0.00001, one-way ANOVA.

**Table 1 biomedicines-11-03141-t001:** Mass and indices of right ventricular hypertrophy in rats on day 21 of the experiment.

Group	*n*	RV Mass, g	RV/Heart	RV/Septum + LV	(RV/Body Mass) × 1000
Control	8	0.167 ± 0.017 *	0.206 ± 0.049 *	0.292 ± 0.045 *	0.522 ± 0.079 *
MCT	8	0.200 ± 0.040	0.231 ± 0.041	0.383 ± 0.064	0.601 ± 0.237
MCT-H_2_	8	0.175 ± 0.025	0.229 ± 0.043	0.368 ± 0.070	0.612 ± 0.245

* Control vs. MCT-Control, MCT-H_2_, *p* < 0.05, one-way ANOVA.

**Table 2 biomedicines-11-03141-t002:** Extracellular connective tissue matrix content in rat lung stroma.

Parameter		Experimental Groups		
		Control (*n* = 6)	MCT-Control (*n* = 6)	MCT-H2 (*n* = 6)
Staining method: Picro Mallory histochemical protocol				
Area of analyzed structures of the airway and respiratory parts of the lung (M, mm^2^) Δ		87.27	102.43	90.02
Area of extracellular matrix of connective tissue	Absolute (M ± m, mm^2^)	13.43 ± 1.2	25.91 ± 3.2 *	17.73 ± 2.1 *^,^**
	Relative (%, M ± m)	15.4 ± 2.2	25.3 ± 2.4 *	19.7 ± 1.9 **

Δ area of lung structures was determined without lumen of airways, alveoli, and vasculature. *—*p* < 0.05 compared to control, **—*p* < 0.05 compared to MCT-Control, one-way ANOVA.

**Table 3 biomedicines-11-03141-t003:** The mean number of lung MCs per 30 fields of view at magnification ×20.

Group	*n*	Giemsa Stain	Tryptase Stain
		MCs, Mean/30 Fields of View at Magnification ×20	
Control	7	26 ± 9 *	23 ± 6 *
MCT-Control	6	52 ± 14	40 ± 13
MCT-H_2_	6	36 ± 13 ^#^	28 ± 10 ^#^

* Control vs. MCT-Control, MCT-H_2_, *p* < 0.05, ^#^ MCT-H_2_ vs. MCT-Control, *p* < 0.05, one-way ANOVA.

## Data Availability

Data are available on request from the authors.

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
