# Peer review of "Hydrogen Inhalation Reduces Lung Inflammation and Blood Pressure in the Experimental Model of Pulmonary Hypertension in Rats"

_biomedicines, 2023, doi:10.3390/biomedicines11123141_

Round 1

Reviewer 1 Report

Comments and Suggestions for Authors

The manuscript entitled "Hydrogen Inhalation Reduces Lung Inflammation and Blood Pressure in Experimental Model of Pulmonary Hypertension in Rats " aim to investigate the effect of breathing 4% hydrogen on a monocrotaline-induced PH rat model, including symptom,  blood pressure, histological changes, and the degree of inflammation. It is an interesting topic, However, we have several concerns: 

1.  Could study group provide 42 days data ? They might find more different effect.

2. Could authors provide echocardiography or RV biomaker data ? ( ex. NT-pro BNP or BNP)

3. English shoud be revised.          

Comments on the Quality of English Language

The language is a little difficult to understand , revised will be better.

Author Response

Reviewer 1

The manuscript entitled "Hydrogen Inhalation Reduces Lung Inflammation and Blood Pressure in Experimental Model of Pulmonary Hypertension in Rats " aim to investigate the effect of breathing 4% hydrogen on a monocrotaline-induced PH rat model, including symptom,  blood pressure, histological changes, and the degree of inflammation. It is an interesting topic, However, we have several concerns: 

Response: Thank you for reviewing our manuscript and for your helpful questions. We responded to your comments below.

  1. Could study group provide 42 days data ? They might find more different effect.

Response: This is a good suggestion. Because the effects of hydrogen are mild and influence many protein expressions, it is possible that greater differences could be observed following chronic or long-term administration of molecular hydrogen. Some clinical studies have indicated that longer duration studies are more effective than shorter duration studies (e.g., LeBaron TW, et al.,  Diabetes Metab Syndr Obes)y. 2020 Mar 24:889-96).

In most experimental studies the authors used  monocrotaline (MCT 60 mg/kg), when modeling MCT-induced pulmonary arterial hypertension (PAH) in rats.  In this case, moderate level of PAH develops within 21 days (3 weeks) , that was confirmed in our previously published paper

Kuropatkina T, et al., Sex-Dependent Protective Effect of Combined Application of Solubilized Ubiquinol and Selenium on Monocrotaline-Induced Pulmonary Hypertension in Wistar Rats. Antioxidants (Basel). 2022 Mar 14;11(3):549.

as well as by others.

Urboniene D, et al., Validation of high-resolution echocardiography and magnetic resonance imaging vs. high-fidelity catheterization in experimental pulmonary hypertension. Am J Physiol Lung Cell Mol Physiol. 2010 Sep;299(3):L401-12.

In 4 weeks (28 days) after MCT administration severe form of PAH developed that was confirmed  by high-resolution echocardiography and magnetic resonance imaging vs. high-fidelity catheterization (Urboniene D  et al., 2010)

Ryan JJ, Marsboom G, Archer SL. Rodent models of group 1 pulmonary hypertension. Handb Exp Pharmacol. 2013;218:105-49; Nogueira-Ferreira R., et al.,. Exploring the monocrotaline animal model for the study of pulmonary arterial hypertension: a network approach. Pulmonary Pharmacology & Therapeutics. 2015; 35: 8-16.

After 6-7  weeks (42-49 days), animals begin to die due to  the development of uncompensated right ventricular heart failure

Buermans HP, et al., Microarray analysis reveals pivotal divergent mRNA expression profiles early in the development of either compensated ventricular hypertrophy or heart failure. Physiol Genomics. 2005;21:314-23.,

which corresponds to the major cause of mortality in people with this disease

Tonelli A.R., Arelli. V, Minai O.A., Newman J., Bair N., Heresi G.A., Dweik R.A. Causes and Circumstances of Death in Pulmonary Arterial Hypertension.  Am J Respir Crit Care Med 2013, Vol 188, Iss. 3, pp 365-369, DOI: 10.1164/rccm.201209-1640OC).

Based on this information, we analysed effects of 4% hydrogen in 21 days after MCT administration.

  1. Could authors provide echocardiography or RV biomaker data ? ( ex. NT-pro BNP or BNP)

Response: In clinical settings the  echocardiography and the indirect biomarkers of PAH severity (BNP or NT-pro BNP) are  widely used to monitor the condition of patients and the effectiveness of treatment in order to avoid more risky direct measurements of pulmonary arterial pressure. In experimental studies on rats echocardiography is used as well for indirect assessment of pulmonary arterial pressure but was validated by the direct measurements of pressure. The advantage of the  echocardiography is the option to perform several measurements in time on one and same animal, like it was done in study of (Urboniene D  et al., 2010).

Urboniene D, Haber I, Fang YH, Thenappan T, Archer SL. Validation of high-resolution echocardiography and magnetic resonance imaging vs. high-fidelity catheterization in experimental pulmonary hypertension. Am J Physiol Lung Cell Mol Physiol. 2010 Sep;299(3):L401-12.

Jasenovec T, Radosinska D, Kollarova M, Vrbjar N, Balis P, Trubacova S, Paulis L, Tothova L, Shawkatova I, Radosinska J. Monocrotaline-Induced Pulmonary Arterial Hypertension and Bosentan Treatment in Rats: Focus on Plasma and Erythrocyte Parameters. Pharmaceuticals (Basel). 2022 Oct 5;15(10):1227. doi: 10.3390/ph15101227. PMID: 36297339; PMCID: PMC9611329.

In our study we preferred to use the direct measurements of pulmonary arterial pressure and the levels of the RV hypertrophy at the end of experiment and to compare those data with the results from animals of the control group.

The levels of NT-pro BNP or BNP on the same MCT-induced PAH model were measured in many studies before

(ex.T L Broderick 1, Y Wang, J Gutkowska, D Wang, M Jankowski Downregulation of  oxytocin receptors in right ventricle of rats with monocrotaline-induced pulmonary hypertension Acta Physiol (Oxf) 2010 Oct;200(2):147-58)

and were not included in protocol of our study.

  1. English shoud be revised.

Response: The manuscript has been edited to improve grammar and readability. Information on the approval of Study Protocol was added. (Lines 92, 93).

Reviewer 2 Report

Comments and Suggestions for Authors

Dear authors, 

thank you for the opportunity to read and review the manuscript.

The topic is very interesting and the paper is well written.

General comments

In recent studies hydrogen administration, inhalational or injection, has shown efficacy on reduction of lung inflammation; this could have an impact on reduction of Pulmonary hypertension and modulation of lung fibrosis; indeed, hydrogen has anti-inflammatory properties, reduces smooth muscle cells proliferation and modulates muscle hypertrophy.

[Fu Z, Zhang J. Molecular hydrogen is a promising therapeutic agent for pulmonary disease. J Zhejiang Univ Sci B. 2022 Feb 15;23(2):102-122. English. doi: 10.1631/jzus.B2100420. PMID: 35187885; PMCID: PMC8861563;

Paulin R, Courboulin A, Meloche J, Mainguy V, Dumas de la Roque E, Saksouk N, Côté J, Provencher S, Sussman MA, Bonnet S. Signal transducers and activators of transcription-3/pim1 axis plays a critical role in the pathogenesis of human pulmonary arterial hypertension. Circulation. 2011 Mar 22;123(11):1205-15. doi: 10.1161/CIRCULATIONAHA.110.963314. Epub 2011 Mar 7. PMID: 21382889; PMCID: PMC3545712].

In COPD patients, inhalational hydrogen seems to reduce inflammation activation, thus ensuring reduction of inflammation-related lung modification [Wang ST, Bao C, He Y, Tian X, Yang Y, Zhang T, Xu KF. Hydrogen gas (XEN) inhalation ameliorates airway inflammation in asthma and COPD patients. QJM. 2020 Dec 1;113(12):870-875. doi: 10.1093/qjmed/hcaa164. PMID: 32407476; PMCID: PMC7785302].

 Specific comments

The sections of the manuscript are clear and well written.

In results section some values are available for the entire study population (n=8 for each groups, Table 1) and some not (e.g. Figure 2, n= 6,5,5; figure 3 n=6,5,6; Table 2 n=6,6,6; Table 3 n=7,6,6), please explain.

Line 207, there is a reason why you choose to use section of left lung? Could this parameter have a role as hydrogen is inhalational?

Can we speculate that the missed effect on PH symptoms of inhalational hydrogen in your study is related to the less efficacy on damage prevention (if compared to the study of Wang et al that used intraperitoneal administration) and in less efficacy of hydrogen therapy when vascular remodeling is already effective and irreversible?

Further studies are needed to understand mechanism of action of hydrogen (also in relation to different routes of administration), physiopatologic mechanism of PH and the associated lung vessel remodeling.

Author Response

Reviewer 2

Dear authors, 

thank you for the opportunity to read and review the manuscript. The topic is very interesting and the paper is well written.

General comments

In recent studies hydrogen administration, inhalational or injection, has shown efficacy on reduction of lung inflammation; this could have an impact on reduction of Pulmonary hypertension and modulation of lung fibrosis; indeed, hydrogen has anti-inflammatory properties, reduces smooth muscle cells proliferation and modulates muscle hypertrophy.

[Fu Z, Zhang J. Molecular hydrogen is a promising therapeutic agent for pulmonary disease. J Zhejiang Univ Sci B. 2022 Feb 15;23(2):102-122. English. doi: 10.1631/jzus.B2100420. PMID: 35187885; PMCID: PMC8861563;

Paulin R, Courboulin A, Meloche J, Mainguy V, Dumas de la Roque E, Saksouk N, Côté J, Provencher S, Sussman MA, Bonnet S. Signal transducers and activators of transcription-3/pim1 axis plays a critical role in the pathogenesis of human pulmonary arterial hypertension. Circulation. 2011 Mar 22;123(11):1205-15. doi: 10.1161/CIRCULATIONAHA.110.963314. Epub 2011 Mar 7. PMID: 21382889; PMCID: PMC3545712].

In COPD patients, inhalational hydrogen seems to reduce inflammation activation, thus ensuring reduction of inflammation-related lung modification [Wang ST, Bao C, He Y, Tian X, Yang Y, Zhang T, Xu KF. Hydrogen gas (XEN) inhalation ameliorates airway inflammation in asthma and COPD patients. QJM. 2020 Dec 1;113(12):870-875. doi: 10.1093/qjmed/hcaa164. PMID: 32407476; PMCID: PMC7785302].

Response: Thank you for reviewing our manuscript and helpful comments. We addressed your specific points below.

 Specific comments

The sections of the manuscript are clear and well written.

In results section some values are available for the entire study population (n=8 for each groups, Table 1) and some not (e.g. Figure 2, n= 6,5,5; figure 3 n=6,5,6; Table 2 n=6,6,6; Table 3 n=7,6,6), please explain.

Response: The reason for this discrepancy is the specificity of the measurement of hemodynamic parameters. The point is that the first step in the procedure was the insertion of an arterial catheter. The mean systemic blood pressure and heart rate were recorded. The next step was to place a catheter in the jugular vein to measure systolic pressure in the right ventricle. This procedure often carries a risk of sudden cardiac arrest, and in this case this parameter cannot be recorded. In addition, animals with pulmonary hypertension have fragile vessels that are easily damaged during catheterization as the procedure is performed blindly. This can lead to pleural hemorrhage and subsequent death. Where a parameter was independent of animal death, such as right ventricular mass or lung mass, it was fixed. Therefore, unfortunately some data could not be measured during the study.

Line 207, there is a reason why you choose to use section of left lung? Could this parameter have a role as hydrogen is inhalational?

Response: The selection of the left lung section for analysis was a deliberate choice to ensure consistency and standardization in our experimental approach. Hydrogen gas, being highly diffusible, has the capability to distribute easily and homogeneously throughout lung tissues. While we specifically focused on the left lung for analytical purposes, we believe that the observed effects can be representative of the broader pulmonary context. The decision to choose a specific lung section was made to minimize variability and enhance the reliability of our experimental results. We appreciate the reviewer's attention to this aspect of our methodology and acknowledge that the homogenous distribution of hydrogen gas is a key consideration in interpreting our findings.

Can we speculate that the missed effect on PH symptoms of inhalational hydrogen in your study is related to the less efficacy on damage prevention (if compared to the study of Wang et al that used intraperitoneal administration) and in less efficacy of hydrogen therapy when vascular remodeling is already effective and irreversible?

Response: It is difficult to say, this may be true. However, as also discussed in lines 338-345, the method of administration may also result in the differences. Intraperitoneal administration provides an intermittent exposure, which may be more effective than the constant exposure used in our study. Additionally, the peak plasma concentration of H2 would be lower in our study vs the study by Wang et al. These factors may also contribute to the attenuated results.

Further studies are needed to understand mechanism of action of hydrogen (also in relation to different routes of administration), physiopatologic mechanism of PH and the associated lung vessel remodeling.

Response: We agree with your assessment. We hope that our second to last paragraph in the discussion section makes this point clear about the need to explore different methods of administration, dosing, frequency, and elucidate the underlying mechanism for hydrogen’s biological effects. This was also modified to include primary target(s) of molecular hydrogen per your suggestion.

Reviewer 3 Report

Comments and Suggestions for Authors

Dear Author,

I have carefully reviewed the manuscript titled "Hydrogen Inhalation Reduces Lung Inflammation and Blood Pressure in an Experimental Model of Pulmonary Hypertension in Rats" and would like to provide my feedback and suggestions for improvement. Overall, the study presents interesting findings regarding the potential benefits of hydrogen inhalation in reducing lung inflammation and blood pressure in a rat model of pulmonary hypertension. I believe that addressing the following points will enhance the clarity, validity, and overall quality of the manuscript.

1. Inclusion of Control-H2 Group:

The manuscript describes three experimental groups: control, MCT-control, and MCT-H2, with the aim of exploring the protective effects of H2 in the MCT-induced pulmonary hypertension rat model. However, since H2 itself is a variable, I recommend adding a control-H2 group to assess the effects of H2 on normal rats. This additional group would allow for a separate analysis of the influence of H2 alone, thereby strictly adhering to the principle of controlling a single variable.

2. Expansion of Analysis on Mast Cell Mediators:

The morphological analysis of lung tissue sections from the MCT-control group provides valuable insights into the inherent pathological changes associated with pulmonary hypertension. The manuscript reports the quantification of mast cells (MCs) using Giemsa staining and highlights the highest MC activity and degranulation around fibrotic areas and narrowed blood vessels. However, the manuscript only presents the release of TGF-β as a mediator by MCs. I recommend expanding the analysis to include other preformed mediators such as histamine, tryptase, and chymase, as well as newly generated mediators including prostaglandin (PG) D2, leukotriene (LT) B4 and LTD4, and a range of cytokines and growth factors such as tumor necrosis factor (TNF)-α, vascular endothelial growth factor (VEGF), granulocyte-macrophage colony-stimulating factor (GM-CSF), IL-10, IL-8, IL-5, IL-3, and IL-1. Additionally, it would be beneficial to provide information on the control, control-H2, and MCT-H2 groups to present a more comprehensive understanding of the results.

3. Organization of Results:

The presentation of results in the manuscript appears somewhat disorganized. To improve clarity and logical flow, I suggest organizing the results section into subheadings that summarize the findings of each part. Each subheading should be followed by the corresponding results presented in a logical order. Furthermore, I recommend providing a comprehensive summary of all experimental results to ensure the comprehensive, systematic, and logical presentation of the findings. Additionally, the placement and clarity of figures 4 and 5 should be improved for better understanding.

I appreciate the valuable research presented in the manuscript and acknowledge its potential contribution to the field. I believe that addressing the above-mentioned points will significantly strengthen the manuscript. Please note that my suggestions are intended to enhance the clarity, validity, and overall quality of the study. Should you require any further clarification or have any questions, please do not hesitate to reach out. Thank you for considering my recommendations.

Comments on the Quality of English Language

The author's writing is fluent and can appropriately express their ideas, but individual image words need to be adjusted

Author Response

Reviewer 3.

Dear Author,

I have carefully reviewed the manuscript titled "Hydrogen Inhalation Reduces Lung Inflammation and Blood Pressure in an Experimental Model of Pulmonary Hypertension in Rats" and would like to provide my feedback and suggestions for improvement. Overall, the study presents interesting findings regarding the potential benefits of hydrogen inhalation in reducing lung inflammation and blood pressure in a rat model of pulmonary hypertension. I believe that addressing the following points will enhance the clarity, validity, and overall quality of the manuscript.

Response: Thank you for reviewing our manuscript and helpful comments. We provided responses to your comments below.

  1. Inclusion of Control-H2Group:

The manuscript describes three experimental groups: control, MCT-control, and MCT-H2, with the aim of exploring the protective effects of H2 in the MCT-induced pulmonary hypertension rat model. However, since H2 itself is a variable, I recommend adding a control-H2 group to assess the effects of H2 on normal rats. This additional group would allow for a separate analysis of the influence of H2 alone, thereby strictly adhering to the principle of controlling a single variable.

Response: We agree that it is important on adding additional controls. Due to shortage of funding and difficulties we opted to first attempt this study with only three groups. There are many studies that have done as you suggested in which H2 was used in the control group, and typically there are no changes. For example, ref 22 (He, B., et al.) used a sham group that was administered oral H2 water. Other studies have also investigated the biological effects of H2 on control cells or animals. Nevertheless, this is still an important area and we hope that we can do this in the future as it is necessary to truly elucidate the biological effects and molecular mechanisms of H2.

  1. Expansion of Analysis on Mast Cell Mediators:

The morphological analysis of lung tissue sections from the MCT-control group provides valuable insights into the inherent pathological changes associated with pulmonary hypertension. The manuscript reports the quantification of mast cells (MCs) using Giemsa staining and highlights the highest MC activity and degranulation around fibrotic areas and narrowed blood vessels. However, the manuscript only presents the release of TGF-β as a mediator by MCs. I recommend expanding the analysis to include other preformed mediators such as histamine, tryptase, and chymase, as well as newly generated mediators including prostaglandin (PG) D2, leukotriene (LT) B4 and LTD4, and a range of cytokines and growth factors such as tumor necrosis factor (TNF)-α, vascular endothelial growth factor (VEGF), granulocyte-macrophage colony-stimulating factor (GM-CSF), IL-10, IL-8, IL-5, IL-3, and IL-1. Additionally, it would be beneficial to provide information on the control, control-H2, and MCT-H2 groups to present a more comprehensive understanding of the results.

Response: We appreciate the insightful suggestion regarding the expansion of our analysis to include a broader range of mediators. We believe we need to also optimize our H2 delivery method perhaps by trying different doses and frequencies. This should be accompanied by measuring more cytokines and mediators as well as an H2 control per your suggestion. We believe that this is important for future work and we have added this comment to the discussion section (Lines 384, 385).

  1. Organization of Results:

The presentation of results in the manuscript appears somewhat disorganized. To improve clarity and logical flow, I suggest organizing the results section into subheadings that summarize the findings of each part. Each subheading should be followed by the corresponding results presented in a logical order. Furthermore, I recommend providing a comprehensive summary of all experimental results to ensure the comprehensive, systematic, and logical presentation of the findings. Additionally, the placement and clarity of figures 4 and 5 should be improved for better understanding.

Response: We appreciate the suggestion on the organization of the manuscript. We attempted different variations and while solving some organizational aspects, it created other issues. The complexity of our results and the interdependence of various aspects creates more challenges in maintaining a logical and seamless narrative. However, per your suggestion we added some subheadings to the manuscript to help orient the reader and guide them through the results section. Additionally, we believe that the results will and organization will be improved by the journal layout.

I appreciate the valuable research presented in the manuscript and acknowledge its potential contribution to the field. I believe that addressing the above-mentioned points will significantly strengthen the manuscript. Please note that my suggestions are intended to enhance the clarity, validity, and overall quality of the study. Should you require any further clarification or have any questions, please do not hesitate to reach out. Thank you for considering my recommendations.

Response: We again appreciate your helpful comments to improve our current manuscript and for future research experiments. We hope our changes satisfy your concerns and that we can focus on incorporating your suggestions in our upcoming work. 

The author's writing is fluent and can appropriately express their ideas, but individual image words need to be adjusted

Response: Thank you for your comment. The manuscript has been revised to improve the English and grammar.

Round 2

Reviewer 1 Report

Comments and Suggestions for Authors

Thank you for response and revised , the manuscript get better after your revised. 

Reviewer 3 Report

Comments and Suggestions for Authors

Dear Author,

I have carefully reviewed the revised version of the manuscript titled "Hydrogen Inhalation Reduces Lung Inflammation and Blood Pressure in an Experimental Model of Pulmonary Hypertension in Rats" and would like to provide my feedback and recommendations. I am pleased to inform you that the manuscript has been significantly improved based on the revisions made, and I recommend its acceptance for publication.

1. Regarding the first point raised in the initial review, your explanation and clarification regarding the absence of a control-H2 group due to funding constraints and difficulties are reasonable and acceptable. I understand the challenges you faced and the limitations that arose from these constraints.

2. With regard to the second point raised, I have observed that you have made appropriate modifications in the revised version of the manuscript. The changes made to expand the analysis on mast cell mediators enhance the comprehensiveness of the study. This revision is deemed acceptable for publication.

3. Concerning the third point raised, I have noticed that you have addressed the issue in the revised manuscript. The inclusion of subheadings have significantly improved the overall quality of the manuscript. Your explanation regarding the organization of the results section and the improved clarity and logical flow is reasonable and acceptable.

I would like to commend you on the revisions made in response to the initial review comments. The manuscript has been substantially strengthened, and the concerns raised have been appropriately addressed. I believe that the study presented in the revised manuscript makes a valuable contribution to the field of pulmonary hypertension research.

Once again, I recommend the acceptance of your manuscript for publication. Thank you for your efforts in addressing the reviewer's comments and for your contribution to the scientific community.

Congratulations on the successful revision, and I look forward to seeing your work published.